# Burnout and Mental Interventions among Youth Athletes: A Systematic Review and Meta-Analysis of the Studies

**DOI:** 10.3390/ijerph191710662

**Published:** 2022-08-26

**Authors:** Dominika Wilczyńska, Wen Qi, José Carlos Jaenes, David Alarcón, María José Arenilla, Mariusz Lipowski

**Affiliations:** 1Physical Education and Social Sciences Department, Gdansk University of Physical Education and Sport, 80-336 Gdansk, Poland; 2Social Anthropology, Basic Psychology and Public Health Department, Pablo de Olavide University, 41013 Seville, Spain

**Keywords:** burnout phenomenon, child and adolescent athletes, psychological intervention, online intervention

## Abstract

(1) Background: The subject of athlete burnout is often discussed among sports psychologists. Interventions to reduce this phenomenon are still under investigation with follow-ups. Thus, the purpose of the current meta-analysis is to examine psychological interventions that was carried out to decrease or eliminate burnout syndrome in young athletes. (2) Methods: Scientific electronic databases were searched, and five published studies published between January and June 2022 that met the criteria were selected. The systematic review and meta-analyses followed the Preferred Reporting Items for Systematic Reviews and Meta-Analyses (PRISMA) guidelines. The Cochrane collaboration tool for assessing the risk of bias was used to assess the studies’ quality. Metafor, a package of the R statistical program, was used to perform the analysis. (3) Results: Cognitive behavioral therapy- and mindfulness-based interventions effectively reduced most dimensions of burnout. Moreover, online interventions were significantly more beneficial in this reduction. (4) Conclusions: There should be more high-quality studies on the effectiveness of psychological interventions in reducing burnout, mainly because it leads to tremendous physical and psychological problems for athletes and their coaches; therefore, it requires particular interventions and prevention strategies.

## 1. Introduction

Sporting activity has the potential to facilitate positive youth development and increase wellbeing through strengthening peer and social relationships, engagement, positive emotions, and sport accomplishments [1,2]. However, sport does not always positively affect children and adolescents. The world’s tendency for early specialization especially leads to dangerous consequences when immature children participate in intensive all-year-round training and competition in a single sport [3]. The lack of diversification and exposure to different sports activities during the developmental periods may underlie the enhanced risk of injuries, stopping motor skill development, psychosocial problems, overtraining syndrome, and a high possibility of burnout and potential dropout from sports [4]. Moreover, children who are specialized early in one selected sporting discipline are deprived of the opportunity to increase their self-esteem by naturally trying different activities and hobbies [5].

The topic of athlete burnout is a frequent element of discussion among sports psychologists and researchers all over the world. The first reports on athlete burnout syndrome were published in the early 1980s, and since then, the issue of athlete burnout has received increasing attention [6]. The first study of burnout in sports was conducted by Caccese and Mayerberg (1984) on female and male athletic college coaches [7]. In the sports science literature, athlete burnout has been defined in various ways. For example, Smiths (1984), in his cognitive–affective model, described athletic burnout as emotional, psychological, and physical exhaustion that causes withdrawal from the activity that was once enjoyable and has become an unpleasant source of stress. The author emphasized that burnout appears when stress-related costs surpass the benefits of sports engagement [8]. On the others hand, Schmidt and Stein (1991), in their sport commitment model, viewed burnout as a reaction to chronic stress when rewards decrease while costs and investments increase, with low satisfaction and resources, and the availability of better alternatives rather than sport [9]. However, one of the theories that have led to a certain consensus among researchers is the concept of athlete burnout as a multidimensional syndrome by Raedeke (1997). It refers to Maslach and Jackson’s (1981) definition of occupational burnout. Through a study conducted on 236 female and male 13–18 age-group swimmers, Raedeke defined burnout through three core dimensions: emotional and physical exhaustion, reduced level of accomplishments, and sport devaluation. Emotional and physical exhaustion is associated with intense training and competition, and thereby the feeling of fatigue. A reduced sense of accomplishment is related to the feeling of inefficacy while athletes cannot achieve personal goals or perform below expectations. Sport devaluation refers to losing interest, a “do not care” attitude, or resentment toward performance and one’s sport. Athletes can hold different levels of these dimensions with raised perceptions on all three determining the burnout syndrome [2,6,10,11]. Athlete burnout can lead to a variety of adverse outcomes. First, affective problems such as low mood and hostility toward the training environment. Second, cognitive issues such as distracted focus, memory, and helplessness. Third, physical aspects, such as fatigue, increased probability of injury, even using doping. Lastly, behavioral issues such as absenteeism and poor sports performance. All these aspects can lead to a final dropout [12]. Consequently, athlete burnout can also lower academic outcomes and performance, leading to disharmonious parent–child and peer relationships [13]. Reducing burnout among young athletes could play an essential role in the general development of young people in society. Feigley (1984) believed that the loss of young elite athletes who retire early, consumed by psychological problems before their peak performance and top form, means the failure to fulfill human potential, and the decline in the quality of national and global sports [14].

Psychological interventions implemented to cure burnout are the techniques of mental training generally used to improve the functioning of athletes depending on age and sports level [15]. Mental training is a set of exercises that, through systematic repetition, lead to the formation and consolidation of the player’s mental qualities and skills, such as the concentration of attention, the self-control of the level of arousal and emotions, and mental resistance in the face of stress [16]. The basic techniques of mental training that control motivational and emotional–cognitive processes include visualization, relaxation, goal setting, and internal dialogue. Those techniques are mainly rooted in cognitive behavioral therapy (CBT), the most commonly used in a sports setting [17].

Athlete burnout, especially among young sport participants, is a severe problem for athletes themselves and their coaches; therefore, it requires particular interventions, prevention strategies, and discussions because of the serious consequences to which it leads. Young athletes suffering from burnout are more likely to withdraw from sporting activity. However, this step to leave the sport is often not a reliever, but may worsen athletes’ mental and moral state. Positive psychology tools and the possibility of using online therapy bring new opportunities to help young people in maintaining their fancy sporting activities [15]. Thus, the purpose of the current meta-analysis is to examine if there are significant differences in selected offline and online psychological interventions in eliminating sports burnout among child and adolescent athletes practicing various sporting disciplines.

## 2. Materials and Methods

### 2.1. Eligibility Criteria

This systematic review and meta-analyses followed the Preferred Reporting Items for Systematic Reviews and Meta-Analyses (PRISMA) guidelines [18]. Published studies were retained for extensive examination if they met the following inclusion criteria: (a) at least one treated and one control group with pre- and post-test measures, RCTs; (b) the participants were young athletes with a maximal age of 25 years; (c) a mental intervention was carried out with outcomes on the basis of burnout data for which an effect size could be calculated. There was no restriction in the language of publication. Across multiple checks, all authors confirmed study eligibility. Three authors completed the final inclusion assessments.

### 2.2. Information Sources

As the main objective of this meta-analysis was to evaluate the efficacy of mental interventions on burnout delivered offline and online, the search was limited to the last 20 years to avoid an over-representation of offline interventions due to the lack of available online technology in previous decades. Between March and June 2022, we searched electronic databases, personal meta-analysis history, and checked with personal research contacts. Electronic database searches occurred in EBSCOhost with the following individual databases selected: All. We also searched Web of Science, PubMed, and Google Scholar. Doctoral and master’s theses were included in the search to cover studies unpublished in journals.

### 2.3. Search Protocol

D.W. and W.Q. initially conducted independent database searches. For the first search, W.Q. used the following search terms: burnout or stress or fatigue or exhaustion AND youth or adolescents or young people or teen or young adults or adolescent or teenager or children AND mental intervention AND athletes or sports or athletics or sport or athlete. In EBSCOhost, both searches used the advanced search option that provided five separate boxes for search terms: Box 1 (burnout or stress or fatigue or exhaustion), Box 2 (youth or adolescents or young people or teen or young adults or adolescent or teenager or children), Box 3 (mental intervention), Box 4 (athletes or sports or athletics or sport or athlete), and Box 5 (TI review).

### 2.4. Data Extraction Process

A data extraction protocol was developed and applied in a systematic and standardized way to each of the studies included in the meta-analysis. The data extraction protocol provided the coding system with the methodological characteristics and results of the reviewed studies, following Lipsey’s guidelines [19]. Checks occurred during the extraction process for potential discrepancies. It was necessary to contact only one study’s authors for missing information about the percentages of females during the data extraction process. In the search for studies, all identified studies were reported in English. Thus, no translation software or searching for a native speaker was needed. This process was carried out independently by two reviewers. Cohen’s kappa coefficient was κ = 0.98 for the categorical variables, and the intraclass correlation (ICC) index for the quantitative variables grouped by domains was above 0.97 in all domains. All data extraction forms are available from the first author.

### 2.5. Data Items

To help in addressing our main aim, the recorded data are as follows: (a) study identification (authors, year of publication, country, and journal name); (b) study design (numbers of participants in the intervention and control groups, type of intervention, type of control group, implementation modality, number of measures, and follow-up time frame); (c) participants (age, percentage of females); (d) intervention characteristics (duration in minutes of each session, duration in weeks of the intervention, number of sessions per week and total number of sessions of the intervention); and (e) results (mean scores, standard deviations, and sample sizes). For all sought information, we coded missing information as not reported.

### 2.6. Risk of Bias

The Cochrane collaboration tool for assessing the risk of bias [20] was used to assess the quality of the studies included in the meta-analysis. It was used with the five RCT studies to determine whether any biases had influenced the true effect of the intervention. Two investigators assessed the risk of bias with the following categories: random-sequence generation, allocation concealment, the blinding of participants and personnel, the blinding of outcome assessment, incomplete outcome data, selective reporting, and other sources of bias. The coded characteristics assessed the quality of the studies included in the meta-analysis and provided an evaluation of the possible biases of the results. They were assessed with four classifications: low risk, high risk, unclear risk, and N/A. Cohen’s kappa coefficient was κ = 0.94.

### 2.7. Effect Size Indices and Data Analysis

The metafor package of the R statistical program [21] and the R Studio program were used to perform the analysis.

Cohen’s d was calculated as measure of effect size for all included studies [22] and the standard error of the calculated effect. A random-effects model was used to pool the effect sizes, and examine the efficacy of the interventions for each of the dimensions of burnout (*reduced sense of accomplishment, emotional and physical exhaustion,* and *devaluation of sports*).

### 2.8. Assessment of Heterogeneity and Publication Bias

Heterogeneity, the extent to which the true effect size of outcome measures varies between studies in the meta-analysis, was quantified using Cochran’s Q statistic and the I^2^ index. For I^2^ estimates, a value of 0% is no heterogeneity, 25% is low heterogeneity, 50% is moderate heterogeneity, and 75% is high heterogeneity [20].

The heterogeneity of effect sizes can be explained with the analysis of moderating variables. Individual random-effect metaregressions evaluated the impact of nine moderating variables that were included in the research: three categorical moderating variables (type of intervention, type of control group, implementation modality) and six continuous moderating variables (age, percentage of females, the duration in minutes of each session, the duration in weeks of the intervention, the number of sessions per week, and the total number of sessions of the intervention).

Publication bias was assessed for each dimension through the visual inspection of funnel plots [23] and Egger’s regression tests [24], which was based on a simple linear regression model; if Z ≥ 1.96 or ≤−1.96, the effect was significant. Funnel-plot asymmetry and significant Egger’s regression test indicated a potential publication bias.

## 3. Results

### 3.1. Study Selection

The EBSCOhost was searched with all databases, which included Medline, Sport Discus, Global Health, and Educational Research Complete. We also searched Web of Science, PubMed, and Google Scholar, and a total of 179 results were returned from 2002–2022. The studies found in the searches were stored, and duplicates were removed. We deleted 46 records because of duplicates or irrelevancy. The abstract of each article was then reviewed, and if appropriate for inclusion in the meta-analysis, the full article was evaluated. We excluded 77 records because they were out of age range or had no reported data about burnout and mental intervention. Next, 28 full-text articles were evaluated for eligibility, and 23 of them were excluded. The reasons included the following: no RCTs or effect sizes could not be extracted. Lastly, five were included. The study selection flowchart is shown in Figure 1. Two reviewers independently selected the five included studies (see Table 1). The degree of agreement calculated using Cohen’s kappa coefficient was κ = 0.96; disagreements were resolved by a third reviewer.

### 3.2. Descriptive Characteristics of the Studies

The studies included in the meta-analysis were published between 2015 and 2020 (see Table 2). The total sample consisted of 430 participants, with a mean percentage of 46.12% of females (33.00%–81.54%). The age range was from 11 to 23 years, with a mean age of 18.82 (2.25).

The control group types were the control group without intervention (41.33%) and waiting list with intervention after study completion (58.67%).

The following techniques were used in the interventions: CBT (76.59%) and MBI (23.41%). Of the total number of participants, 43.41% participated in offline implementation modality, and 56.59% were in online implementation modality.

In all five studies, pretest and post-test measures were taken; in one of them, data were collected in the middle of the intervention (midtest), and in two, data were collected several months after of the intervention (follow-up).
ijerph-19-10662-t002_Table 2Table 2Details of the studies included in the systematic review.Authors% FemaleAgeInternetExperimental GroupControl GroupDurationMeasuresInstrumentRisk of BiasLangan et al. (2015) [25]
15.18 (1.29)R= 11–17OfflineCBT*n*= 41Waiting list*n* = 46Weeks = 12Session (min) = 702ABQ− + − + − ? +Luzzeri (2020) [26]81.5420.09 (1.25)R= 18–23OnlineCBT*n*= 31Control*n* = 34Weeks = 2Session (min) = 204ABQ− ? ? ? + ? +Moen and Wells (2016) [27]3318.5OfflineMBI*n* = 25Control*n* = 32Weeks = 12Session (min) = 122ABQ+ + ? ? + ? +Moen et al. (2015) [28]5118.5R= 16–20OfflineMBI*n* = 23Control*n* = 27Weeks = 12Session (min) = 202ABQ+ + ? ? + ? +Ofoegbu et al. (2020) [29]35.0920.38 (3.11)R= 11–21OnlineCBT*n* = 85Waiting list*n* = 86Weeks = 12Session (min) = 903ABQ− − − − − ? ?Notes. For risk of bias, − = low risk of bias, + = high risk of bias, ? = unclear risk of bias on the following indices: random sequence generation, allocation concealment, blinding of participants and personnel, blinding of outcome assessment, incomplete outcome data, selective reporting and other sources of bias. ABQ = Athlete Burnout Questionnaire [30,31]. CBT = cognitive behavioral therapy. MBI = mindfulness-based intervention.

### 3.3. Bias Risk Assessment

The risk of bias was low in 71% of studies for random-sequence generation, 12% for allocation concealment, 33% for the blinding of participants and personnel, 12% for the blinding of outcome assessment, 33% for incomplete outcome data, and 0% for selective reporting and other sources of bias. The risk of bias was high in 29% of studies for random-sequence generation, 50% for allocation concealment, 21% for the blinding of outcome assessment, 67% for incomplete outcome data, 0% for blinding of participants and personnel and selective reporting, and 88% for other sources of bias. In the other cases, the risk of bias was unclear (see Figure 2). See Table 2 for the individual studies.

### 3.4. Effectiveness of Interventions

The five RCT studies with 2 types of intervention, 2 types of control group, and 2 implementation modalities, 15 measures, and 3 dimensions of burnout were analyzed. Significant effect sizes were obtained for all dimensions. Figure 3, Figure 4 and Figure 5 show the forest plot for each dimension that explains a diagrammatic synthesis of the studies’ effects, confidence interval, and summary random effect (RE) for the metanalytical random-effect model.

Thus, we obtained moderately significant effect sizes in *reduced sense of accomplishment* (*d* = −0.74, *SE* = 0.31, *p* < 0.05). *Reduced sense of accomplishment* was significantly decreased with the intervention’s programs (95%CI: −1.34, −0.14). The *Q*-statistic and the *I*^2^ index showed that heterogeneity was moderate in *reduced sense of accomplishment* (*Q* (*df* = 4) = 8.84, *p* = 0.07; *I*^2^ = 54.73%).

**Figure 3 ijerph-19-10662-f003:**
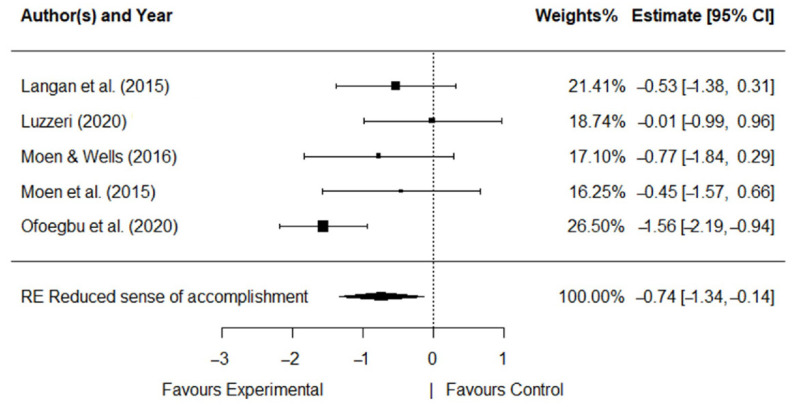
Forest plot for *reduced sense of accomplishment* [25,26,27,28,29].

A large effect was observed for *emotional and physical exhaustion* (*d* = −0.87, *SE* = 0.20, *p* < 0.001). *Emotional and physical exhaustion* was significantly reduced in the intervention programs (95%CI: −1.25, −0.48). The *Q*-statistic and the *I*^2^ index showed that heterogeneity was absent for *emotional and physical exhaustion* (*Q* (*df* = 4) = 3.25, *p* = 0.52; *I*^2^ = 0.00%).

**Figure 4 ijerph-19-10662-f004:**
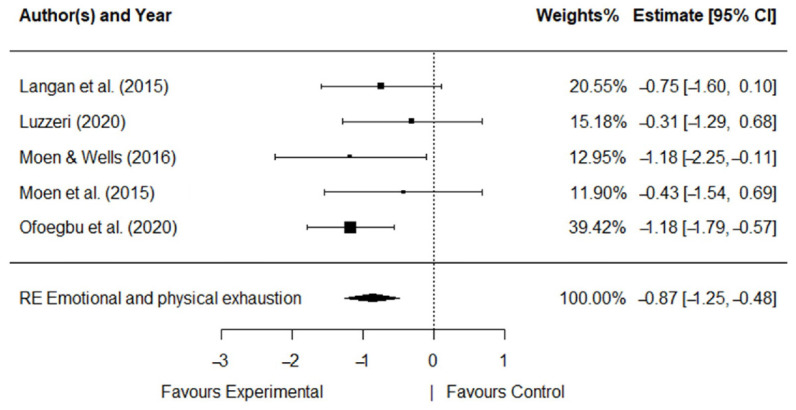
Forest plot for *emotional and physical exhaustion* [25,26,27,28,29].

A moderate effect was observed for *devaluation of sports* (*d* = −0.77, *SE* = 0.33, *p* < 0.05). *Devaluation of sports* was significantly decreased with the intervention’s programs (95%CI: −1.43, −0.11). The *Q*-statistic and the *I*^2^ index showed that heterogeneity was significant for *devaluation of sports* (*Q* (*df* = 4) = 10.56, *p* < 0.05; *I*^2^ = 62.11%).

**Figure 5 ijerph-19-10662-f005:**
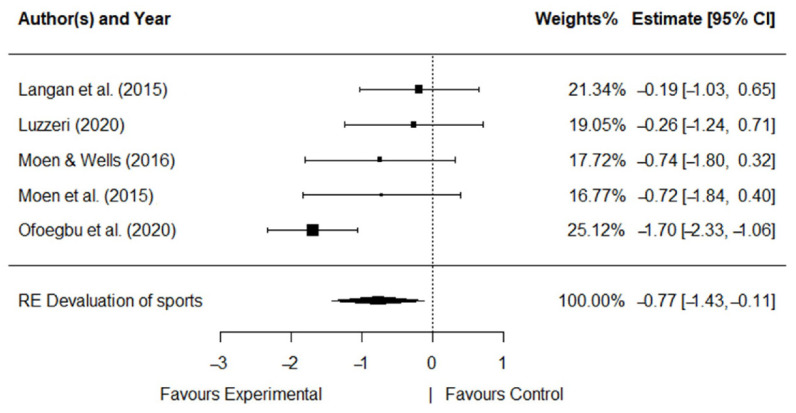
Forest plot for *devaluation of sports* [25,26,27,28,29].

Publication bias was calculated using Egger’s regression test: *reduced sense of accomplishment* (*Z* = 2.45, *p* < 0.05); *emotional and physical exhaustion* (*Z* = 1.18, *p* = 0.24); *devaluation of sports* (*Z* = 1.39, *p* = 0.16). Figure 6, Figure 7 and Figure 8 indicate funnel plots for each dimension.

### 3.5. Influence of Moderating Variables on Efficacy

Table 3 shows the moderating variables that predicted variation in aggregate effect sizes for the *reduced sense of accomplishment* dimension. No significant differences were found between the different types of intervention or according to the form of implementation. However, significant differences were found between the control group types, finding more of an effect when the control group was the waiting-list one instead of the control group. No significant differences were found according to age, the duration of interventions in weeks, the number of sessions per week, or the total number of sessions. However, significant differences were found according to the percentage of females and according to the duration of sessions in minutes.

Table 4 shows the moderating variables that predicted variation in aggregate effect sizes for the *emotional and physical exhaustion* dimension. Significant differences were found between the different types of intervention, proving to be a more effective type of CBT intervention than MBI. Significant differences between the control group types were found, finding more effect when the control group was waiting list instead of control. Significant differences were also found according to the form of implementation; online interventions were more effective than offline interventions. No significant differences were found according to age, percentage of female participants, the duration of the sessions in minutes, duration of the interventions in weeks, number of sessions per week or total number of sessions.

Table 5 explains the moderating variables that predict variation in aggregate effect sizes for the *devaluation of sports* dimension. No significant differences were found between the different types of intervention or types of control group. Significant differences were found depending on the form of implementation; online interventions were more effective than offline interventions. No significant differences were found according to age, the percentage of female participants, the duration of the sessions in minutes, the duration of the interventions in weeks, the number of sessions per week, or the total number of sessions.

## 4. Discussion

The current meta-analysis focused on studies (RCT) on psychological interventions that aimed to reduce young athletes’ burnout published between January 2002 and 10 June 2022. We considered both online and traditional (offline) methods of psychological interventions, and inclusion and exclusion criteria. Lastly, 5 RCT studies with 2 types of intervention, 2 types of control group, and 2 implementation modalities, 15 measures, and 3 dimensions of burnout were selected for description. The meta-analysis compared the effectiveness of cognitive behavioral therapy (CBT) and mindfulness-based intervention (MBI) interventions in decreasing burnout, such as a reduced sense of accomplishment, the devaluation of sport, and emotional and physical exhaustion. While considering the effectiveness of psychological intervention in general, we obtained a moderately significant effect size in reduced sense of accomplishment and devaluation of sports. The intervention’s programs significantly decreased both parameters of burnout. However, the CBT interventions were more effective in reducing emotional and physical exhaustion than MBI was. As previously mentioned, CBT techniques are most commonly used in sports; however, mindfulness techniques that focus on presence may be beneficial [17,32]. The systematic review on depressive symptoms and burnout in football players emphasized the positive effects of both CBT and MBI in preventing and curing mental problems [33]. Breslin et al. (2022) underlined in a systematic review of the interventions to increase the awareness of mental health and wellbeing in athletes and coaches that there is a considerable need for longitudinal studies with larger samples of males and females, and validated measurement tools [34]. Observations show that, in some cases, athletes are even more prone to mental disorders when going through different adversities such as burnout, injury, and long competition periods when being away from friends and family. Furthermore, sports coaches, who are expected to play the role of the gatekeepers of athletes’ mental health, suffer similar mental disorders as those of athletes [35].

While searching for the predictors of the changes in the three analyzed burnout dimensions, the reduced sense of accomplishment was positively changed, while the control group character was waiting list instead of just control, and the percentage of females among respondents and duration of sessions in minutes. If the respondents were females, the positive effects of the intervention were significantly higher, which is related to other studies proving that female athletes are significantly more open to mental consultation and less stigmatizing to the psychological help itself. Moreover, females are more involved in mental training and convinced of its usefulness than male athletes are [36,37]. Similarly, the emotional and physical exhaustion dimension was more significantly reduced in the waiting-list control group. Online interventions were interestingly more effective in reducing the emotional and physical exhaustion, and the devaluation of sports dimensions compared to offline, traditional face-to-face interventions. Openness to online help is seen in sports settings since the lockdown caused by the COVID-19 pandemic. The use of interventions based on modern technologies seems to be increasingly effective. For example, many sports associations have introduced especially prepared online mental-health interventions for athletes during confinement. Furthermore, studies revealed that athletes who had received such professional psychological help coped better with psychological stressors, showing more improved wellbeing than those who did not participate in such training. Online interventions give new opportunities and allow for helping athletes wherever they are [38].

The present study has the major strength that all the reviewed studies used the same measure of burnout, so the effect of the interventions on each of the subdimensions of the questionnaire was analyzed. However, this meta-analysis has a few limitations due to the small number of studies that met the inclusion criteria. The total participant size among all the reviewed studies was low, and caution is needed with the obtained results. In addition, the combination of moderating variables is not always complete; for instance, the reviewed studies that were provided in online format performed a CBT intervention; thus, the effect of an MBI intervention delivered online was not analyzed. Another limitation is that the effects of the intervention were only analyzed among young athletic participants. Future research could expand the number of analyzed studies by comparing the effect of the intervention in young and adult athletes. Because only five studies met the criteria for the meta-analysis, sensitivity analysis excluding some studies to assess the stability of the results could not be performed. Future reviews and meta-analyses with broader inclusion criteria might evaluate the effects of alternative assumptions in the research questions. However, this analysis could be beneficial for sports clubs and all the professionals who work with athletes to implement mindfulness or cognitive behavioral tools in the training process.

## 5. Conclusions

Five studies were selected for the analysis. The meta-analysis revealed that both CBT and MBI interventions were effective forms of reducing most burnout dimensions. Moreover, online interventions seemed to be more beneficial for young athletes’ burnout healing, showing that new forms of therapy could be more beneficial. The current meta-analysis and previous systematic reviews emphasize a strong need for more research on psychological interventions’ effectiveness in reducing burnout among young athletes, mainly because the phenomenon of athlete burnout still requires more observations and measurements.

## Figures and Tables

**Figure 1 ijerph-19-10662-f001:**
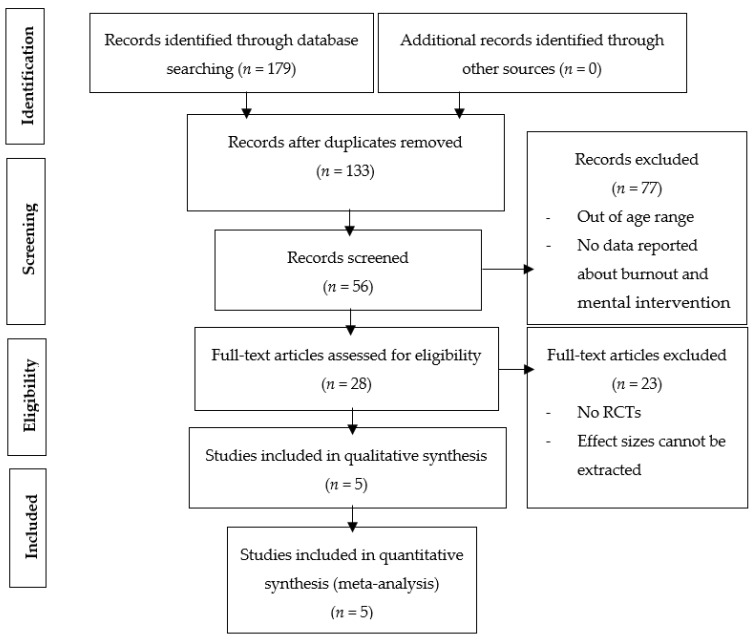
PRISMA flowchart for the identification of the included studies.

**Figure 2 ijerph-19-10662-f002:**
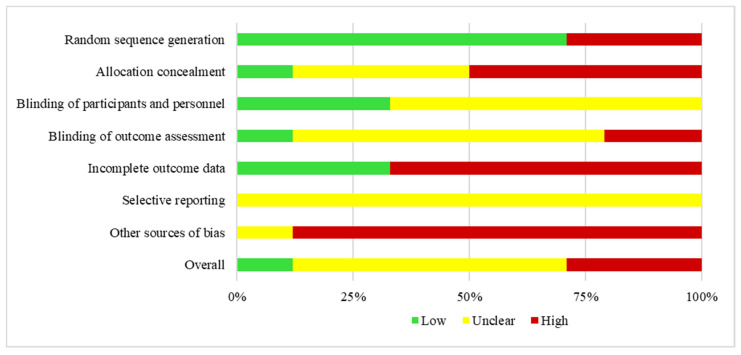
Risk of bias in the studies included in the meta-analysis.

**Figure 6 ijerph-19-10662-f006:**
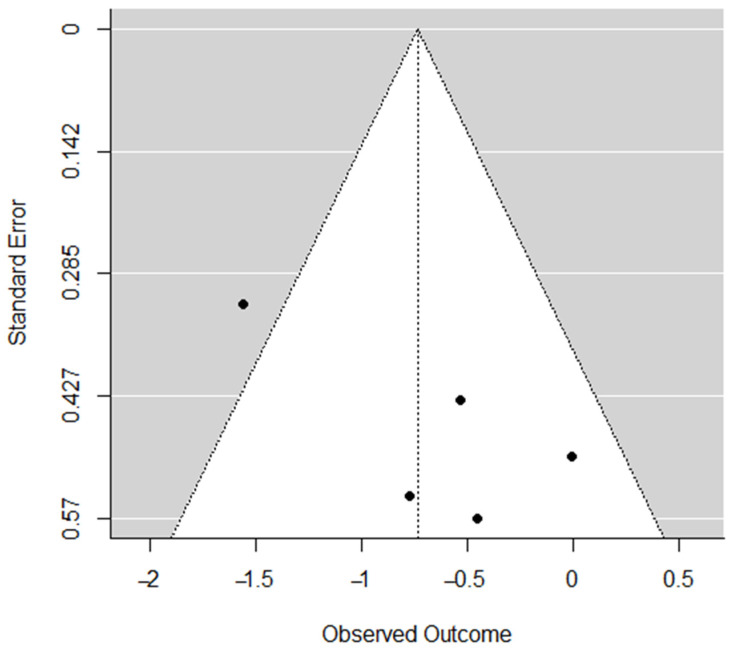
Funnel plot for *reduced sense of accomplishment*.

**Figure 7 ijerph-19-10662-f007:**
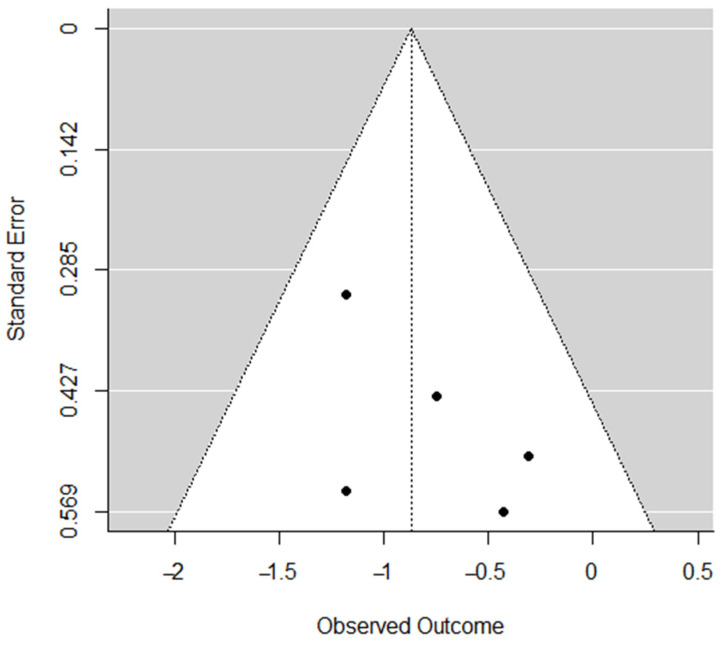
Funnel plot for *emotional and physical exhaustion*.

**Figure 8 ijerph-19-10662-f008:**
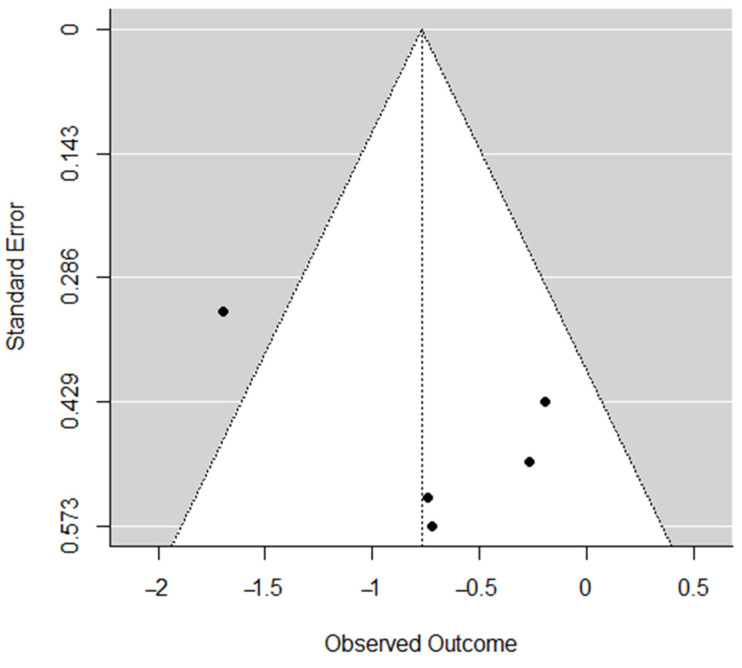
Funnel plot for *devaluation of sports*.

**Table 1 ijerph-19-10662-t001:** Empirical studies of burnout.

Title	Database	Author	Year	Journal	Aim	Participants	Methods	Instrument	Result
Testing the Effects of a Self-Determination Theory-Based Intervention with Youth Gaelic Football Coaches on Athlete Motivation and Burnout.	Web of Science	Langan et al. [25]	2015	The Sport Psychologist	To test the effects of a self-determination theory-based intervention on athlete motivation and burnout. To examine the feasibility and acceptability of the intervention.	The sample (*n* = 87) comprised players between 11 and 17 years of age (experimental: *M* = 15.02, *SD* = 1.62; control: *M* = 15.34, *SD* = 0.96).	This study is a cluster randomized controlled trial (RCT) analyzing change in player motivation and burnout as a result of their coach participating in a 12-week SDT-based intervention. We randomly assigned coaches (*n* = 6) and their teams to an experimental or control group (i.e., usual practice) (*n* = 3 each group)	Behavioral Regulation in Sport Questionnaire (BRSQ). Athlete Burnout Questionnaire (ABQ). Perceived Environmental Supportive Questionnaire (PESQ). Controlling Coach Behavior Scale (CCBS).	The findings demonstrated the feasibility and acceptability of implementing a self-determination-theory-based intervention in the coaching domain. In addition, this study demonstrated favorable trends in the quality of player motivation and burnout symptoms as a result of an SDT-based intervention.
The effects of a meaning-oriented onlinw writing intervention on commitment, stress, and burnout in collegiate athletes.	ProQuest	Luzzeri et al. [26]	2020	Dissertation	To evaluate the efficacy of an online writing intervention in treating burnout in collegiate athletes. Specifically, it was hypothesized that increases in the presence of meaning in sport through writing would lead to decreases in burnout symptoms.	The sample consisted of 65 student athletes from various NCAA programs across the United States. The final sample included male (*n* = 12) and female (*n* = 53) athletes ranging from 18 to 23 years of age (*M* = 20.09, *SD* = 1.25).	Screening involved 425 NCAA collegiate athletes from a variety of sports, with 157 qualified participants, 86 agreeing to participate, for a total of 65 participants completing the intervention. The online intervention included six sessions to be completed over the course of two weeks (i.e., three sessions each week). The study was composed of four distinct phases, namely, screening, group assignment, intervention or control, and follow up.	Meaning in Sport Questionnaire (MSQ). Sport Commitment Questionnaire-2 (SCQ-2). Satisfaction With Sport Scale (SWSS). A modified version the Perceived Stress Scale (PSS). Athlete Burnout Questionnaire (ABQ).	Results from a series of repeated-measure ANCOVA showed marginal improvements in constrained commitment and presence of meaning in sport for the intervention group, with no other changes in burnout or related constructs. Manipulation check results using LIWC software suggested that the writing intervention elicited the content they were designed for. Findings are discussed in light of new research on meaning in sport, theoretical approaches of athlete burnout, and future research directions in both domains.
Can the Attention Training Technique Reduce Burnout in Junior Elite Athletes?	Semantic Scholar	Moen and Wells [27]	2016	International Journal of Coaching Science	To investigate the effects of the attention-training technique on junior elite athletes’ level of burnout. To examine an ancillary question of whether the technique impact on mindfulness.	This study comprised 78 Norwegian junior elite athletes. The ATT condition consisted of 27 athletes, while the control condition consisted of 51. A gender breakdown of the participants showed that 67% of the entire sample were male and 33% were female. Their average age was 18.5 years. Out of the 78 elite athletes who participated in the project at the pretest, 57 athletes participated at the post-test after 12 weeks (25 in the experimental group, and 32 in the control group). This gave a response rate of 73%.	After assignment to a group, a pretest was administrated through an online questionnaire that measured the psychological variables in this study. Then, an ATT program was administrated for a period of 12 weeks. Athletes participated in the post-test after 12 weeks. The ATT used in this study was a audio training which has the goal of enhancing mental control and flexibility.	Athlete Burnout Questionnaire (ABQ). Mindful Awareness Attention Scale (MAAS).	The results showed that there was a significant decrease in burnout among the ATTgroup, but not the control group. In the ATT group, but not the control, mindfulness increased.
The Effects from Mindfulness Training on Norwegian Junior elite Athletes in Sport.	Web of Science	Moen et al. [28]	2015	International Journal of Applied Sports Sciences	To investigate the effects of a mindfulness intervention on perceived stress, perceived performance in school and sports, and athlete burnout among junior elite athletes in sports.	This study comprised 77 Norwegian junior athletesin sports. The gender breakdown of the subjects was 49% men and 51% women. Their average age was 18½ years old (ranging from 16 to 20). Out of the 77 athletes who participated in the project at the pretest, 50 athletes participated after 12 weeks, which gave a response rate of 65% (23 in the experimental group, and 27 in the control group).	The experimental design of this study was a pretest–post-test control group. After the junior elite athletes had been randomly assigned into either the experimental or the control group, a pretest was administrated. The junior athletes then answered an online questionnaire that measured the psychological variables in this study. Then, a mindfulness program was administrated for a period of 12 weeks.	Mindful Awareness Attention Scale (MAAS). Perceived Stress Scale (PSS). Athlete SatisfactionQuestionnaire (ASQ). Athlete Burnout Questionnaire (ABQ).	As hypothesized, significant effects were found from the mindfulness intervention on athlete burnout. There were no significant effects on perceived stress, perceived performance in school and sports.
Effect of digital storytelling intervention on burnout thoughts of adolescent.	Web of Science	Ofoegbu et al. [29]	2020	Medicine	To ascertain the effect of a digital storytelling intervention on the burnout thoughts of adolescent athletes with disabilities.	This study involved 171 adolescent para-athletes preparing for various local and international competitions in Southeastern Nigeria. Intervention group (*n* = 85; *M* age = 20.18, *SD* age = 3.15) and waitlisted control group (*n* = 86; *M* age = 20.57, *SD* age = 3.07)	This study is a randomized controlled trial involving a total of 171 adolescent athletes with disabilities who showed a high degree of burnout symptoms. These adolescent athletes were randomly assigned to either an intervention group or a waitlisted control group. The treatment intervention for the adolescent athletes was digital stories that had been created on the basis of the framework of rational emotive behavior therapy (REBT). The questionnaire was used for gathering data at three different times (baseline, post-test, and follow-up). Data were analyzed using repeated-measure analysis of variance at a significance level of 0.05.	Athlete Burnout Questionnaire (ABQ).	Results showed that the digital storytelling intervention based on REBT significantly reduced burnout thoughts among disabled adolescent athletes in the intervention group compared to athletes in the waitlisted control group as measured with the Athlete Burnout Questionnaire. Additionally, follow-up evaluation showed that the decrease in burnout scores was maintained by those athletes in the digital storytelling intervention.

**Table 3 ijerph-19-10662-t003:** Moderating variables for the efficacy of interventions on *reduced sense of accomplishment*.

Moderating Variables	*k*	*d*	95% CI	*Qm*	*Qe*	*p*
Intervention type ^a^	5			4.60	8.31 *	
CBT		−0.78	[−1.60, 0.04]			0.06
MBI		−0.62	[−1.74, 0.51]			0.28
Control type ^a^	5			10.77 **	4.75	
Control		−0.39	[−1.12, 0.33]			0.29
Waiting list		−1.13	[−1.84, −0.42]			<0.01
Internet ^a^	5			5.31	7.05	
Offline		−0.58	[−1.44, 0.27]			0.18
Online		−0.91	[−1.85, 0.04]			0.06
Age ^b^	5	−0.09	[−0.42, 0.25]	0.26	7.11	0.61
Female percentage ^b^	4	0.03	[0.00, 0.05]	4.63 *	2.31	<0.05
Duration in minutes ^b^	5	−0.01	[−0.03, −0.00]	4.29 *	3.42	<0.05
Duration in weeks ^b^	5	−0.09	[−0.23, 0.05]	1.68	5.35	0.20
Sessions per week ^b^	5	0.11	[−0.21, 0.44]	0.47	7.00	0.49
Total sessions ^b^	5	0.00	[−0.03, 0.28]	0.01	8.68 *	0.94

Notes. *k* = number of studies; *d* = mean effect size; CI = confidence interval; *Qm* = test of moderators; *Qe* = test for residual heterogeneity. * *p* < 0.05; ** *p* < 0.01. ^a^ Categorical moderating variables. ^b^ Continuous moderating variables.

**Table 4 ijerph-19-10662-t004:** Moderating variables for the efficacy of interventions on *emotional and physical exhaustion*.

Moderating Variables	*k*	*d*	95% CI	*Qm*	*Qe*	*p*
Intervention type ^a^	5			17.38 ***	3.23	
CBT		−0.87	[−1.34, 0.40]			<0.001
MBI		−0.82	[−1.61, 0.03]			<0.05
Control type ^a^	5			20.59 ***	2.23	
Control		−0.62	[−1.23, −0.02]			<0.05
Waiting list		−1.03	[−1.53, −0.53]			<0.001
Internet ^a^	5			18.44 ***	3.11	
Offline		−0.79	[−1.37, −0.21]			<0.01
Online		−0.93	[−1.47, −0.39]			<0.001
Age ^b^	5	−0.04	[−0.24, 0.15]	0.20	3.05	0.66
Female percentage ^b^	4	0.02	[−0.00, 0.04]	2.63	0.52	0.10
Duration in minutes ^b^	5	−0.01	[−0.02, 0.01]	1.16	2.09	0.28
Duration in weeks ^b^	5	−0.07	[−0.17, 0.04]	1.47	1.78	0.22
Sessions per week ^b^	5	0.06	[−0.15, 0.28]	0.35	2.91	0.56
Total sessions ^b^	5	−0.00	[−0.02, 0.02]	0.00	3.25	0.97

Notes. *k* = number of studies; *d* = mean effect size; CI = confidence interval; *Qm* = Test of Moderators; *Qe* = Test for Residual Heterogeneity. *** *p* < 0.001. ^a^ Categorical moderating variables. ^b^ Continuous moderating variables.

**Table 5 ijerph-19-10662-t005:** Moderating variables for the efficacy of interventions on *devaluation of sports*.

Moderating Variables	*k*	*d*	95% CI	*Qm*	*Qe*	*p*
Intervention type ^a^	5			4.05	10.31 *	
CBT		−0.77	[−1.69, 0.15]			0.10
MBI		−0.73	[−1.97, 0.51]			0.25
Control type ^a^	5			5.19	8.37 *	
Control		−0.57	[−1.50, 0.36]			0.23
Waiting list		−1.00	[−2.01, 0.01]			0.05
Internet ^a^	5			6.94 *	6.67	
Offline		−0.52	[−1.35, 0.31]			0.22
Online		−1.09	[−2.01, −0.17]			<0.05
Age ^b^	5	−0.20	[−0.48, 0.09]	1.81	5.42	0.18
Female percentage ^b^	4	0.02	[−0.01, 0.05]	2.54	2.80	0.11
Duration in minutes ^b^	5	−0.01	[−0.03, 0.01]	1.08	6.44	0.30
Duration in weeks ^b^	5	−0.06	[−0.23, 0.11]	0.50	8.61 *	0.48
Sessions per week ^b^	5	0.05	[−0.34, 0.43]	0.06	9.86 *	0.81
Total sessions ^b^	5	−0.00	[−0.03, 0.03]	0.01	10.54 *	0.93

Notes. *k* = number of studies; *d* = mean effect size; CI = confidence interval; *Qm* = test of moderators; *Qe* = test for residual heterogeneity. * *p* < 0.05. ^a^ Categorical moderating variables. ^b^ Continuous moderating variables.

## Data Availability

The data is available by contacting co-author David Alarcón: dalarub@upo.es.

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
