# Peer review of "Burnout and Mental Interventions among Youth Athletes: A Systematic Review and Meta-Analysis of the Studies"

_ijerph, 2022, doi:10.3390/ijerph191710662_

Round 1
Reviewer 1 Report
Title:
Please, specify in the title this research is a systematic review with meta-analysis.
Introduction
In the introduction, several sentences are not justified with references. Please, review and justify.
Explain more consistently “Why it is important to do this review and meta”.
Besides the aim, provide the research questions.
Materials and Methods
The authors affirm: “This systematic review and meta-analyses followed the Preferred Reporting Items 98 for Systematic Reviews and Meta-Analyses (PRISMA) guidelines.”, provide the reference.
Inclusion criteria:
“(b) published prior to June 10, 2022”; why reason this is a eligibility criterion? I don’t understand it.
“(c) the participants were young athletes”; Could you provide a range of age?
What is the primary variable? The authors must indicate the variables of the study in this section.
Exclusion criteria.
Did the authors not include exclusion criteria?
Please state the reason why you focused on studies no older than 20 years.
2.2. Information sources 107
Specify the databases includes (each one). And gray literature?
Explain the following section: Data extraction and management
Explain the following section: Risk-of-bias assessment in the included studies. You must explain how was carried out the study of risk of bias and management (remember this is the method section)
Explain the following section: Treatment effect measures, to carry out the meta-analysis.
Explain the following section: “Assessment of heterogeneity and reporting bias”. Explain heterogeneity analysis, funnel plot, etc.
Explain the following section: “Sensitivity analysis”
Results.
The authors must reorganize (some sections of the methods are results) and complete the section. For example, the content of Figure 1 and Table 1 are the results.
Figure 1: The authors must include the reasons to eliminate the studies (articles) in Figure 1 (following PRISMA).
Discussion
The first paragraph must include a summary of the main results.
Include, Overall completeness and applicability of evidence of this study.
As well, analyze the quality of the evidence included in this meta-analysis and analysis of potential biases in the review process (about your own meta-analysis).
Include also, Implications for practice and Implications for research.
Author Response
We greatly appreciate the work of the reviewer. Thank you for your effort and valuable comments, which help us grow in our work and develop our abilities. Our responses to the reviewer's comments are in the Cover letter and revision sheet.
Warm regards

Reviewer 2 Report
I found the article comprehensive and interesting. Much is said about the benefits of sport on emotional development, but it is also necessary to pay attention to the risks that young athletes may experience, due to the demands of competition, early specialisation, etc.
This article provides a rigorous review and analysis of the interventions that have been used to prevent burnout in young athletes, the selected papers are homogeneous and consistent and the results of the interventions have always been evaluated in the same way, which facilitates the comparison of the effects of the interventions. The authors have been very careful in the presentation of the methodology used, they follow the Prisma methodology and go into great detail on aspects such as the mediating variables that can influence the results of the interventions used. There is only one aspect where I would ask them to make some changes and that is in the inclusion and exclusion criteria of the selected papers. The authors comment that the age of the participants is an exclusion criterion for many studies, but it is not clear to me exactly what age range they have focused on. This should be explicitly stated among the inclusion and exclusion criteria in figure 1. Also, there is a typo in line 321 where they write ad instead of "and".
Author Response

(The authors gave the same response as above.)

Round 2
Reviewer 1 Report
Congratulations!